# South China sea issue and Southeast Asian countries' perception of China's image: An empirical study based on GDELT big data

Zibo Wei[1,2], Xuan Chen[3]*, Genli Tang[4], Yishuai Xie[2]

1 Department of International Affairs, School of International Studies, University Utara Malaysia, Kedah, Malaysia, 2 Office of Human Resources, Sichuan Vocational College of Health and Rehabilitation, Zigong, China, 3 School of Politics and Administration, China West Normal university, Nanchong, China, 4 Business School, Xuzhou University of Technology, Xuzhou, China

* jhnfpt@163.com

## Abstract

How does the South China Sea issue affect China's image in Southeast Asian countries? Has it diminished Southeast Asian countries' perception of China? Based on the Global Database of Events, Language, and Tone (GDELT) from 2010 to 2024, this study empirically examines the impact of the South China Sea issue on Southeast Asian countries' perception of China's image using a panel multiple linear regression model. The empirical results show a significant positive correlation between the South China Sea issue and Southeast Asian countries' perception of China's image. Specifically, the positive impact of verbal events, material events, and the scale of events related to the South China Sea significantly enhances the positive evaluation and recognition of China among Southeast Asian countries. Additionally, GDP plays a moderating role in this process, weakening the positive impact of verbal events, factual events, and the scale of South China Sea events on China's positive evaluation and recognition. In Southeast Asian countries with territorial disputes and sovereignty conflicts with China, the influence of factual events and the scale of South China Sea events on the perception of China's image is more pronounced. In contrast, in Southeast Asian countries without such disputes, verbal events have a more significant impact on their perception of China's image.

## 1 Introduction

The South China Sea issue, as a longstanding and prominent topic in international relations, has been a significant variable in the relationship between Southeast Asian countries and China. The South China Sea region not only possesses abundant natural resources but also serves as a crucial international shipping route, involving multiple interests such as national sovereignty, freedom of navigation, economic development, resource exploitation, and military security. In recent years, overlapping

**Data availability statement:** All relevant data are within the manuscript and its Supporting Information files.

**Funding:** The author(s) received no specific funding for this work.

**Competing interests:** The authors have declared that no competing interests exist.

territorial and maritime claims between China and several Southeast Asian countries have made the South China Sea issue a major concern for regional security and stability. Against this backdrop, Southeast Asian countries' perception of China's image has been significantly affected. This perception not only influences bilateral relations but also plays a key role in shaping the overall China policy of ASEAN. Moreover, with the increasing "intervention" of the United States, the South China Sea disputes have intensified, leading to challenges and reshaping of China's image in Southeast Asia. Therefore, exploring how the South China Sea issue affects Southeast Asian countries' perception of China is crucial for understanding regional political dynamics and promoting the peaceful resolution of disputes.

This paper, based on the vast data from the Global Database of Events, Language, and Tone (GDELT), employs an empirical research approach to analyze the portrayal of the South China Sea issue in Southeast Asian media and public opinion, as well as its impact on China's national image. As a global data platform, GDELT covers news events and media reports from almost every country and region worldwide, providing comprehensive and detailed data support on specific issues [1]. Through data mining and analysis, we can better understand the dissemination patterns of the South China Sea issue in Southeast Asian countries and the social responses it triggers. This, in turn, reveals the impact of the South China Sea issue on China's national image, offering valuable insights for both policy formulation and academic research [2].

## 2  Southeast Asian countries' perception of China's image

Image is a product constructed through cognition [3]. The perception of a particular country by the cognitive subject forms the national image [4]. In essence, a national image is a highly generalized and stable subjective cognition, shaped by traditional stereotypes, an evolving cognitive process, and clear stances [5,6]. Among the many factors influencing national image, the cognitive subject's personality traits, behavioral attitudes, values, and recognition of a country's political ecosystem, economic environment, cultural heritage, and institutional cooperation have profound and lasting impacts on the cognitive outcome [7,8]. On a macro level, factors such as political systems, economic foundations, cultural beliefs, power structures, relative strength, and the attitudes of ruling elites between countries can influence how one country's image is perceived, designed, and modified through the "cognitive lens [9–11]." Referring to Chen (2022) [12], The economic influence China in Southeast Asia is significant, with 59.5% of respondents viewing China as the most influential economy, far exceeding the United States' 14.3%, thanks to trade and investment cooperation. This shows that the volume of trade has a potential impact on the image of China as perceived by Southeast Asian countries.

### 2.1  Image perception of China in Southeast Asian countries

Theoretically, the individual attributes of Southeast Asian citizens—such as education level, ethnic beliefs, value systems, psychological characteristics, historical traditions, and cognitive frameworks—along with their environment, as well as the differences

between Southeast Asian countries and China in terms of politics, economics, culture, and social identity, all influence Southeast Asian countries' perception of China's national image. From the perspective of the state system, factors such as the positive interactions between Southeast Asian countries and China [13,14] (including the Belt and Road Initiative and the China-ASEAN Free Trade Area), the conflicts and tensions between China and certain Southeast Asian countries over the South China Sea issue, Southeast Asian countries' perceptions of China's rise, their sense of identity and centrality within ASEAN, their understanding of the intensifying strategic competition between China and the U.S., as well as their strategic demands for economic development and security assurance—all contribute to shaping and forming Southeast Asian countries' national image of China [15–17].

Before empirically testing the impact of the South China Sea issue on Southeast Asian countries' perception of China, this paper first utilizes the sentiment and tone index (AvgTone) and the Goldstein scale from the Global Database of Events, Language, and Tone (GDELT) as independent variables to assess and measure the overall perception and attitude of Southeast Asian countries toward China. GDELT is the world's largest open-access database of political and news media events, designed to monitor and analyze global news media events. Currently, GDELT collects information at a rate of 100,000–150,000 new events per day from media sources worldwide. The database categorizes more than 300 types of events into four major categories: verbal cooperation, material cooperation, verbal conflict, and material conflict. Each event contains 58 fields, including the time of the event, the countries of the actors and targets, names, organizations, event categories, the nature of the event, the level of impact on participants, frequency of media mentions, sentiment tone, geographic information, event date, and source of information, among others. GDELT updates interaction events between countries in real time approximately every 15 minutes, encoding and scoring both verbal (e.g., public statements, cooperation intentions, warnings, resistance) and material actions (e.g., tangible cooperation, conflicts). The database uses two methods of scoring: the tone index reflects the sentiment and attitude of media when reporting the event, while the Goldstein scale measures the intensity of the event. The tone index ranges from -100–100, where lower values indicate a more negative perception and attitude toward a country, and higher values suggest a more positive perception. The Goldstein scale ranges from -10–10, with lower values indicating a higher level of conflict between two countries and higher values reflecting greater cooperation.

Fig 1 uses the monthly weighted average of the sentiment index (AvgTone) of Southeast Asian countries toward China, as recorded in the GDELT database, to measure their perception and attitude towards China's national image. It also employs the monthly weighted average of the Goldstein scale to assess the bilateral political relations between Southeast Asia and China (To eliminate the interference of subjective factors such as media, this paper uses the number of times each event is mentioned by news media (NumMentions) as a weight to calculate the weighted average of AvgTone and Goldstein. Data source: The GDELT Project, "GDELT 1.0 Event Database," https://www.gdeltproject.org). As shown in Fig 1, during the sample period (2010–2024), (The data collection period is from January 1, 2010, to July 31, 2024.) there is a clear trend of increasing negative perception of China among Southeast Asian countries, with February 2015 marking the first occurrence of more negative perceptions than positive ones. In January 2013, the Philippines formally initiated arbitration proceedings against China under the United Nations Convention on the Law of the Sea (UNCLOS). The political relationship between ASEAN countries and China also reached its lowest point of -2.14 in March 2013. From a temporal perspective, January 2013, when the Philippines initiated the "South China Sea Arbitration," became a pivotal event, leading to a series of conflicts and tensions between China and the Philippines, China and Vietnam, and China and ASEAN over the South China Sea issue. This event notably shifted Southeast Asian countries' perceptions of China. Vietnam and Cambodia hold different views of China due to their unique geopolitical and historical backgrounds, with Vietnam holding a more negative view of China due to its territorial disputes in the South China Sea and its long history of antagonism with China. This is evidenced by the frequent coverage of the South China Sea disputes in the Vietnamese media and the strong public reaction to China's actions in the region [18].

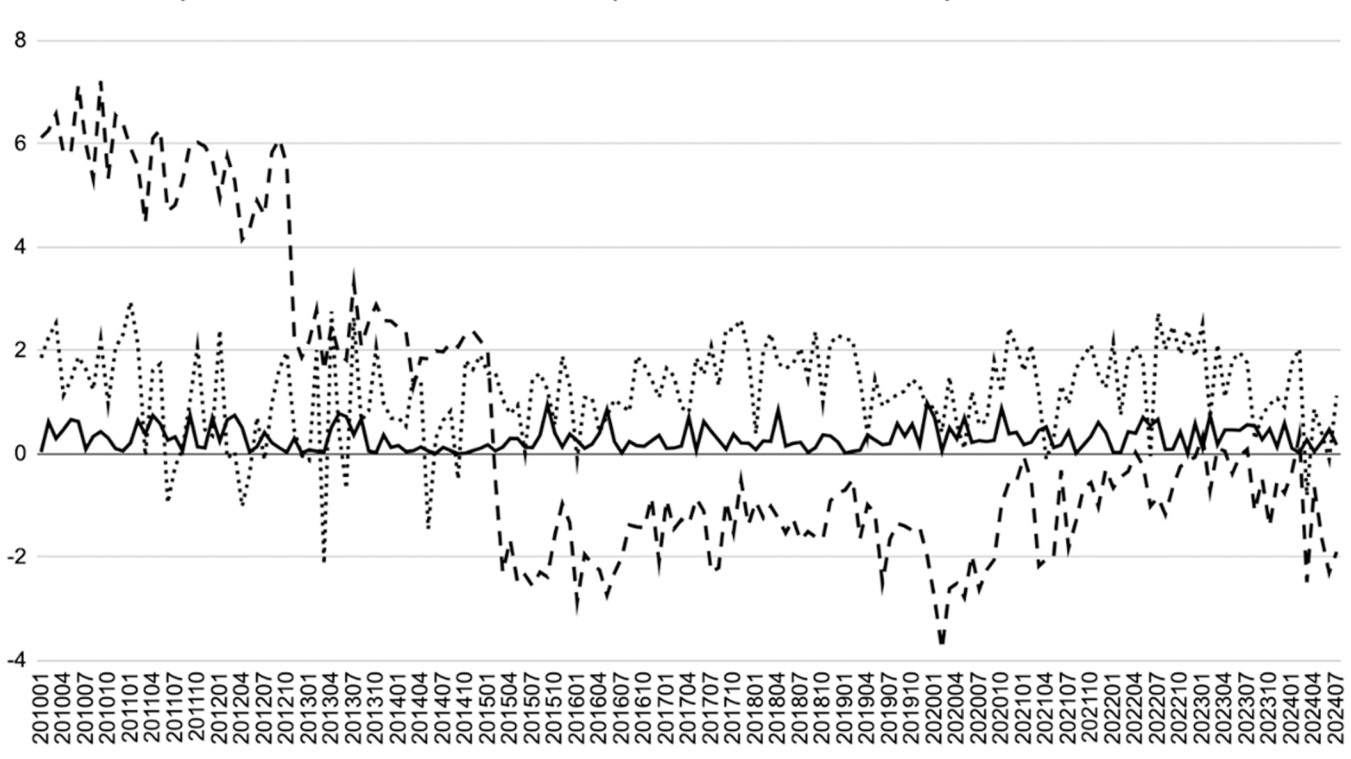

**Fig 1. The Relationship between Southeast Asian Countries and China and Their Perception of China.**

Cambodia, on the other hand, maintains close economic ties with China and has received significant Chinese investment in areas such as infrastructure, resulting in a relatively positive perception of China. The Cambodian government's neutral stance on the South China Sea contrasts sharply with Vietnam in terms of the intensity of media and public discussion [19].

By examining data from 2010 to 2024, it is observed that Laos, Cambodia, and Singapore have significantly more positive perceptions of China compared to other Southeast Asian countries, while the Philippines and Vietnam exhibit the highest levels of negative perception. It is noteworthy that, although there are certain differences in perceptions of China among Southeast Asian countries, overall, the perceptions have remained relatively stable and convergent, with limited fluctuations. The perception differences of 11 Southeast Asian countries towards China, measured by the monthly average of the AvgTone from the GDELT database, slightly increased from 0.036 in January 2010 to 0.473 in July 2024, reflecting a change of only 0.133 over nearly 15 years (Perception difference is defined as the absolute value of the difference between the monthly weighted average of AvgTone of Southeast Asian countries towards China and that of China towards the respective country for a specific event. Data source: The GDELT Project, "GDELT 1.0 Event Database."). A possible theoretical explanation is that the formation of ASEAN has fostered a stronger sense of collective identity among Southeast Asian countries, and institutional constraints within ASEAN have contributed to narrowing the perception differences towards China among its member states [20,21].

### 2.2 Cognitive dissonance, misinterpretation of signals, and Southeast Asian countries' negative perception of China

Among the various factors influencing a country's negative perception of another, in addition to inherent stereotypes, identity affiliations, and long-established beliefs and cultural differences—factors that are stable and long-term—cognitive

dissonance, misinterpretation of signals, and misunderstandings of emotional expressions can also lead to perceptual biases and misjudgments in a country's perception of another [22].

In recent years, Southeast Asian countries have exhibited an intensifying negative perception of China, primarily due to the increased frequency and intensity of conflictual events in their interactions with China, which has reinforced their misperceptions and negative attitudes towards China. China has dealt with maritime disputes by demonstrating a willingness to cooperate (e.g., joint training) and by taking a tough stance (e.g., in response to arms sales to Taiwan), sending dual signals through military, diplomatic and economic means [23]. According to statistics from the Global Database of Events, Language, and Tone (GDELT), between 2010 and 2024, a total of 535,268 interaction events occurred between Southeast Asian countries and China. Of these, 405,507 were cooperative events (including 341,794 verbal cooperation events and 63,713 material cooperation events), while 129,761 were conflictual events (including 83,570 verbal conflict events and 46,191 material conflict events), accounting for 75.76% and 24.24% of the total number of events, respectively. In terms of event nature, the proportion of cooperative events decreased from its peak of 94.95% in 2013 to 80.99% in 2023, while the proportion of conflictual events increased from 14.98% in 2010 to 26.90% in 2023 (In the GDELT database, Quad class represents the types of events: 1 and 3 represent verbal cooperation and verbal conflict, respectively; 2 and 4 represent material cooperation and material conflict, respectively. Data source: The GDELT Project, "GDELT 1.0 Event Database.").

In terms of event intensity, the increase in the intensity of conflictual events, especially material conflict events, is positively correlated with the rise in negative perceptions of China among Southeast Asian countries (Fig 2). It is the increase in both conflictual events and the degree of conflict that has significantly undermined the positive effects of cooperative

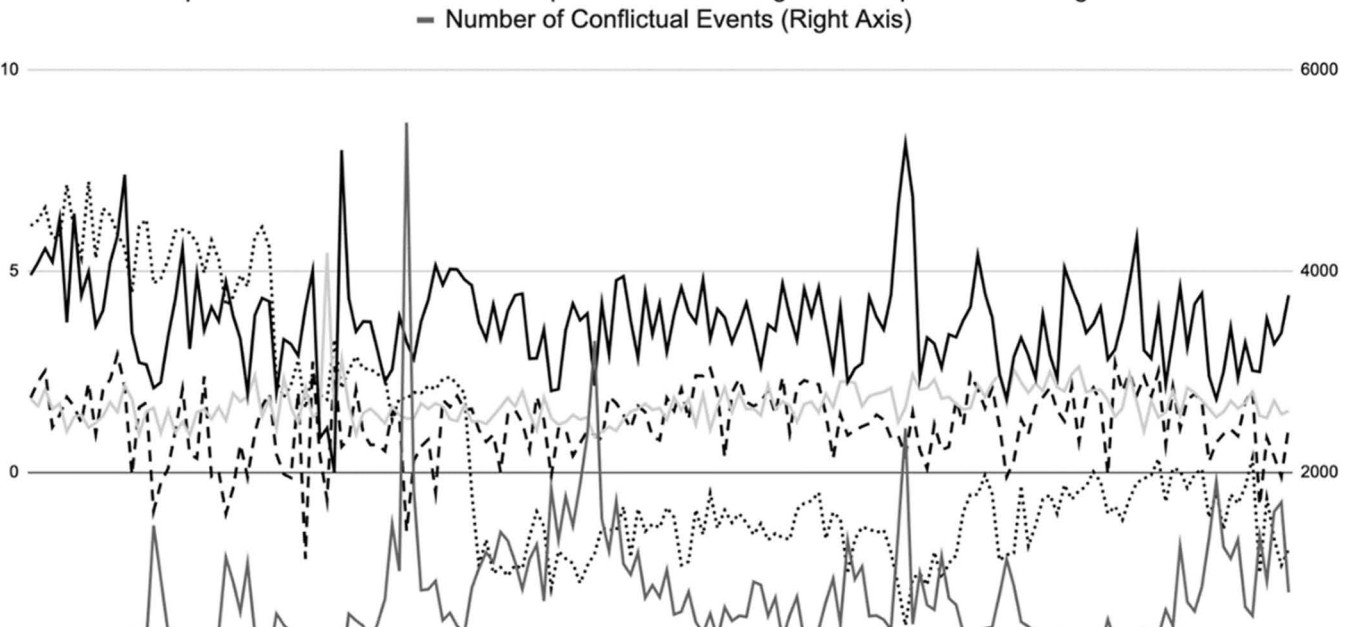

**Fig 2. The Impact of Material Conflicts on Southeast Asian Countries' Perception of China.**

events, thereby reinforcing cognitive dissonance and negative perceptions of China among Southeast Asian countries. The fundamental reason for this cognitive dissonance and negative perception lies in the "imbalance" in Southeast Asian countries' perception of China [24].

### 2.3 Systemic factors and Southeast Asian countries' negative perception of China

Systemic factors, particularly shifts in the relative power between major powers, not only influence actors' perception of systemic shocks and their understanding of external challenges but also shape a country's strategies in response to systemic pressures. The ratio of China's economic, military, and technological strength relative to the United States increased from 39.60%, 10.70%, and 32.76% in 2010 to 65.40%, 25.45%, and 40.41% in 2023, respectively (The ratio of economic strength refers to the GDP ratio, with data sourced from UNCTAD STAT, http://unctadstat.unctad.org/EN/; the ratio of military strength refers to the military expenditure ratio, with data sourced from the SIPRI Military Expenditure Database, https://www.sipri.org/databases/milex; and the ratio of technological strength refers to the ratio of total factor productivity (TFP), with data sourced from PWT 10.0, https://www.rug.nl/ggdc/productivity/pwt/?lang=en.). At the same time, the China-U.S. relations index, as measured by the Tsinghua University's assessment of great power relations, declined from 1.2 in May 2010 to -8.3 in April 2023 (When the score falls within the range of -9 to -6, the bilateral relationship is classified as "antagonistic"; a score between -6 and -3 indicates a "tense" relationship; between -3 and 0, the relationship is considered "discordant"; between 0 and 3, it is "neutral"; between 3 and 6, it is "friendly"; and between 6 and 9, the relationship is considered "amicable." "Antagonistic" refers to a state where the nature of the relationship between the two countries is hostile, and they openly regard each other as strategic adversaries, although no direct large-scale military conflicts occur. See Yan, X. T., et al. (2010). *Overview of China's Foreign Relations 1950–2005: Quantitative Measurement of China's Relations with Major Powers*. Beijing: Higher Education Press, p. 1. Data source: http://www.tuiir.tsinghua.edu.cn/info/1145/6075.htm.). Since turning negative for the first time in July 2016, the relationship score between China and the U.S. has not returned to positive territory (Fig 3). From a realist perspective, the rise in the United States' negative perception of China, the continuous decline in strategic relations between China and the U.S., and the narrowing gap in relative power between the two countries are closely interconnected [25].

Empirical research indicates that changes in the power dynamics between China and the United States, along with the intensification of strategic competition, have not only heightened the risk perception and hostile narrative of the United States towards China but also increased the misperceptions of China's rise among countries and regions, including Southeast Asia. This shift has complicated their decision-making on whether to align with, hedge against, or balance between the United States and China. In other words, the changes in the national power of China and the United States, adjustments in the power structure, and the escalation of strategic competition have a significant impact on the perception structure and narrative approach of the United States and its allied countries towards China's national image. Simultaneously, these factors also profoundly influence the strategic preferences of neighboring regions and countries, including Southeast Asian states, in choosing between the United States and China [22].

According to the data measured by the monthly average of the Goldstein scale from the Global Database of Events, Language, and Tone (GDELT), there is no clear intention among Southeast Asian countries to "take sides" in response to the current stage of strategic competition between China and the United States. The relationship between China and the United States is tending towards friendship and cooperation. However, due to historical reasons, political and cultural factors, there are also differences and conflicts. On the whole, there is more cooperation than conflict, and common interests outweigh mutual differences [26]. Even countries like the Philippines and Vietnam, which have serious disputes and conflicts with China over the South China Sea issue, have not fully aligned themselves with the United States in their diplomatic actions and foreign strategies (In the diplomatic relations between China and the United States, the equidistance engagement metric is defined as the difference between a country's monthly average Goldstein score for events related to China and its monthly average score for events related to the United States. A larger

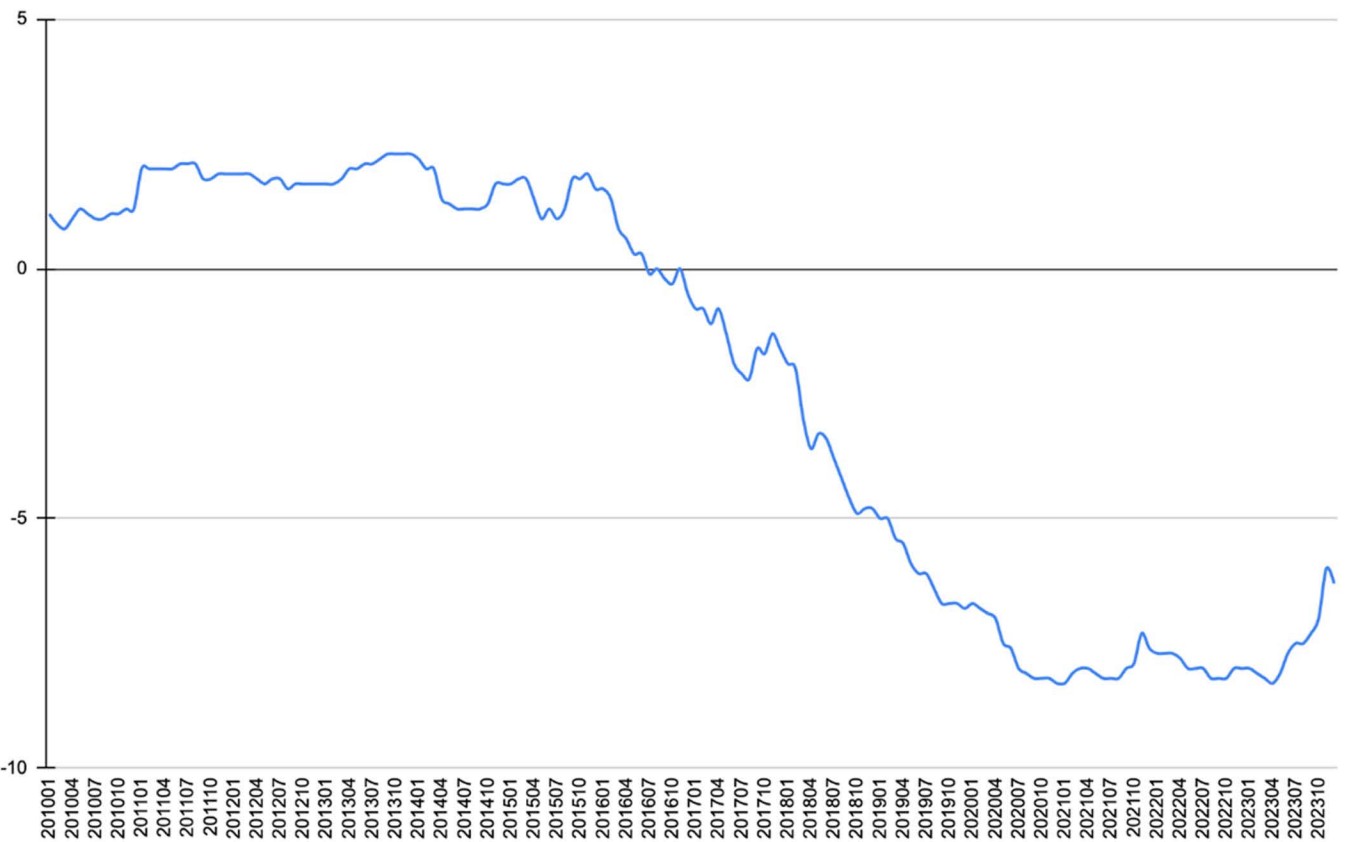

**Fig 3. Changes in the China-U.S. Relationship Score Since 2010.** (Compiled based on the China-Great Power Relations Score Table released by Tsinghua University, this table accurately reflects the changes in China's relations with other countries. As of now, the data on the Tsinghua University website is available up to December 2023. Data source: http://www.tuiir.tsinghua.edu.cn/info/1145/6075.htm).

value indicates a diplomatic preference toward China, while a smaller value suggests a diplomatic inclination toward the United States. On the other hand, the hedging tendency index is quantified by calculating the square of the difference between a country's monthly average Goldstein score for events related to China and that for the United States. A smaller value indicates a stronger diplomatic hedging tendency between China and the United States, whereas a larger value suggests a greater inclination toward balancing or aligning with one of the two. Data source: The GDELT Project, "GDELT 1.0 Event Database."). Most countries have chosen to maintain close military and security ties with the United States while simultaneously maintaining friendly economic, trade, and diplomatic relations with China. China faces the need to adjust and upgrade its industrial structure in dealing with financial crises, and it is well-positioned to conduct industrial transfer on a gradient basis. The manufacturing equipment that China phases out in its own industrial restructuring and upgrading process is still in demand in the less developed countries of Southeast Asia. This equipment can be invested in these countries for joint ventures to produce and sell locally [27]. However, based on the monthly average tone data from the GDELT, it is evident that the negative perception of China among Southeast Asian countries is steadily rising. This trend appears to be correlated with the shifts in the power structure between China and the United States, the narrowing gap in relative strength, the intensification of strategic competition, and the increasingly negative perception of China in the United States.

## 3  Has the 'South China Sea Issue' diminished Southeast Asian Countries' perception of China's image?

"Has the 'South China Sea Issue' Diminished Southeast Asian Countries' Perception of China's Image? Based on global event big data from 2010 to 2024, this paper will employ multiple linear regression analysis to empirically test whether the 'South China Sea Issue' has diminished Southeast Asian countries' perception of China's image."

### 3.1  Data and variables

**3.1.1  Dependent variable.**  AvgTone is an emotional sentiment metric in the GDELT database, used to measure the tone of news coverage, with values ranging from -1 to +1, where negative values indicate negative sentiment and positive values indicate positive sentiment. Users can access news data containing AvgTone through the GDELT API or database, then calculate and analyze the AvgTone values of multiple events or reports to reveal overall emotional trends. Additionally, AvgTone can be combined with factors like time, location, or topic to visualize sentiment fluctuations, and is widely used in sentiment analysis and public opinion research.

This paper uses the annual average of the AvgTone index from the Global Database of Events, Language, and Tone (GDELT) as the dependent variable for China's national image. The AvgTone index is commonly used to measure the positive or negative perception of one country toward another. The index ranges from -100–100, with higher values indicating a more positive perception of China's national image, while lower values indicate a more negative perception. The sample includes 11 Southeast Asian countries over a period of 15 years (2010–2024).

**3.1.2  Key explanatory variable.**  The GDELT database, by categorizing and quantifying global news events, facilitates the quantification of the "South China Sea issue." Therefore, this study employs three key explanatory variables—scale of South China Sea events (AVGgoldstein), impact of verbal events (VC), and impact of material events (MC)—to measure the influence of the South China Sea issue on the changes in bilateral relations between China and Southeast Asian countries.

Sister city relationships are theoretically expected to promote cultural and social ties between countries, thereby influencing national image. However, such relationships typically involve exchanges at the local government or societal level, whereas national image formation is primarily shaped by high-level diplomatic policies, geopolitical events, and sovereignty disputes. When it comes to issues related to national security and territorial conflicts, cultural exchanges at the local level may have limited impact on shaping public perceptions of a country as a whole.

Although sister city programs facilitate economic, cultural, and educational cooperation at the local level, their influence remains relatively limited and is often not widely disseminated through media coverage. Additionally, public awareness of these cooperative initiatives tends to be low, making their role in shaping national image less significant compared to more influential diplomatic or economic events. This study focuses on how the South China Sea dispute affects China's national image in Southeast Asia. In this context, the South China Sea issue dominates media discourse and public perception, potentially overshadowing less politically salient forms of cooperation such as sister city relationships, making their impact less visible in the statistical model.

**3.1.3  Control variable.**  To better reveal the causal relationship between the "South China Sea issue" and China's national image, this paper incorporates variables such as a country's economic, military, and diplomatic ties with China into the model to control for the influence of bilateral relations between Southeast Asian countries and China on its national image (This study uses investment dependency and export volume to measure a country's economic ties with China, with data sourced from the United Nations Comtrade Database, the *Statistical Bulletin of China's Outward Foreign Direct Investment*, and the UNCTAD STATA Data Center. The study uses data from the Swedish International Peace Research Institute's "SIPRI Arms Transfers Database" on the value of Chinese arms imported by foreign countries to assess the military relationship between the countries. Additionally, the number of sister cities and geographical distance are used to evaluate diplomatic relations, thereby measuring the strategic depth and levels of a country's relationship with China. Data sources include the International Sister City Query System and the CEPII database). Although existing

theories and empirical studies have shown that close economic ties and deepening cooperative interactions with China do not necessarily translate into positive perceptions of China—and may sometimes even lead to friction or conflict— overall, the close political, economic, military, and diplomatic connections between China and Southeast Asian countries, especially in the context of China's continuous goodwill gestures, increasing tangible cooperation outcomes, and deepening bilateral cooperation, suggest that the proper management of the "South China Sea issue" could positively influence Southeast Asian countries' perceptions of China.

**3.1.4 Moderating variable.** To better identify the role of relevant variables in analyzing the relationship between the South China Sea issue and Southeast Asian countries' perception of China's image, this study includes a country's GDP as a moderating variable in the model. GDP is not only a crucial indicator of a country's economic strength and development level but also plays a significant role in international relations and diplomatic behavior. By selecting GDP as a moderating variable, this study aims to explore how the level of economic development moderates political and diplomatic perceptions. Specifically, a higher GDP may make a country more sensitive and alert to external conflicts, thus leading to different response patterns in its perception of China's image. Therefore, the moderating role of GDP helps to more precisely understand the mechanism by which economic factors influence Southeast Asian countries' perceptions of China in the context of the South China Sea issue. The detailed description of the indicator system is shown in Table 1.

## 3.2 Econometric model specification

**3.2.1 Model construction.** The article takes the perception of China's image by various Southeast Asian countries as the dependent variable, examining the impact of verbal event influence, material event influence, and the scale of South China Sea incidents on this perception. Additionally, export volume, the quantity of weapons imported from China, investment dependence, sister city relationships, and bilateral geographic distance are selected as control variables. GDP is introduced as a moderating variable for testing moderation effects through regression analysis. The baseline regression model is as follows:

$$AvgtoneY_{it} = \beta_0 + \beta_1 VC_{it} + \beta_2 EX_{it} + \beta_3 MIL_{it} + \beta_4 INV_{it} + \beta_5 FCITY_{it} + \beta_6 DIS_{it} + \mu_{it}$$

$$AvgtoneY_{it} = \beta_0 + \beta_1 MC_{it} + \beta_2 EX_{it} + \beta_3 MIL_{it} + \beta_4 INV_{it} + \beta_5 FCITY_{it} + \beta_6 DIS_{it} + \mu_{it}$$

$$AvgtoneY_{it} = \beta_0 + \beta_1 AVGgoldstein_{it} + \beta_2 EX_{it} + \beta_3 MIL_{it} + \beta_4 INV_{it} + \beta_5 FCITY_{it} + \beta_6 DIS_{it} + \mu_{it}$$

The moderation effect model is as follows:

$$AvgtoneY_{it} = \beta_0 + \beta_1 X_{it} + \beta_2 GDP_{it} + \beta_3 X * GDP_{it} + \beta_4 EX_{it} + \beta_5 MIL_{it} \\ + \beta_6 INV_{it} + \beta_7 FCITY_{it} + \beta_8 DIS_{it} + \mu_{it}$$

Here, AvgtoneY represents the perception of China's image by Southeast Asian countries. The explanatory variables include: impact of verbal events (VC), impact of material events (MC), and the influence of the scale of South China Sea events (AVGgoldstein). EX denotes export scale, MIL represents the number of weapons imported from China, INV signifies investment dependency, FCITY indicates the number of sister cities, and DIS refers to bilateral geographic distance. $\beta_0$ represents the constant term, while $\beta_i$ are the regression coefficients of each variable. Table 2 presents the descriptive statistical results of each variable.

Based on the table above, the standard deviations of variables such as the perception of China's image, the impact of verbal events, the impact of material events, and the impact of the South China Sea event scale all exceed 1. This indicates that there are significant differences among Southeast Asian countries in their perception of China's image, as well as in the influence of verbal events, material events, and the scale of South China Sea events.

**Table 1. Explanation of the indicator system.**

| Category | Variable Name | Definition | Data Source | Notes |
|---|---|---|---|---|
| Dependent Variable | AvgTone | Perception of China's Image | GDELT Database | 1. Annual processing;<br>2. Values calculated using Python |
| Explanatory Variable | AVGgold-stein | Impact of the Scale of South China Sea Events | GDELT Database | 1. Annual processing;<br>2. Values calculated using Python<br>3. The impact of the scale of South China Sea events is used to assess the cooperative or conflictual tendencies of these events. A higher AVGgoldstein value indicates a greater degree of peace, signifying that the scale of South China Sea events has a significant positive impact. Conversely, a lower AVGgoldstein value indicates a higher degree of conflict, signifying that the scale of South China Sea events has a significant negative impact |
| | VC | Impact of Verbal Events | GDELT Database | 1. Annual processing<br>2. Values calculated using Python<br>3. Number of verbal cooperation events minus the number of verbal conflict events, representing the positive impact of verbal events |
| | MC | Impact of Material Events | GDELT Database | 1. Annual processing<br>2. Values calculated using Python<br>3. Number of material cooperation events minus the number of material conflict events, representing the positive impact of material events. |
| Control Variable | Invest | Investment Dependence | China's Statistical Bulletin of Outward Foreign Direct Investment | 1. Annual processing<br>2. Analysis based on China's outward investment flow data |
| | Ex | Export Value | Comtrade Database | Annual processing |
| | Mil | Quantity of weapons imported from China | SIPRI Arms Transfers Database | Annual processing |
| | Fcity | Sister City | International Sister Cities Query System | Annual processing |
| | Dis | Geographical distance between Southeast Asian countries and China | CEPII Database | The geographical distance between the capitals is used as the geographical distance between the country and China. |
| Moderating Variable | GDP | Gross Domestic Product | WDI Database | Annual processing |

**3.2.2 Baseline regression.** A further regression analysis was conducted to examine the impact of various factors on image perception. The Hausman test results for each model show that the p-values are all 0, indicating that each model is suitable for fixed effects regression. The results are shown in Table 3.

The results of the baseline regression indicate that all models passed the F-test at the 1% significance level, suggesting that the overall regression performance of each model is satisfactory.

The regression coefficient between the impact of verbal events and the perception of China's image is 0.0076, which passed the T-test. This indicates a positive correlation between the impact of verbal events and the perception of China's image. In other words, the higher the difference between the number of verbal cooperation events and verbal conflict events, the more positive the perception of China's image by various countries. The impact of verbal events has a positive effect on the perception of China's national image.

**Table 2. Descriptive statistics of variables.**

| Variable | Obs | Mean | Std.Dev. | Min | Max |
|---|---|---|---|---|---|
| AvgtoneY | 154 | 0.9229595 | 2.998374 | -4.538916 | 7.178274 |
| VC | 131 | 46.84733 | 143.5447 | -94 | 1136 |
| MC | 131 | -1.664122 | 12.78196 | -64 | 41 |
| AVGgoldstein | 131 | 0.3681624 | 2.144541 | -7.146448 | 7 |
| EX | 154 | 3.1291 | 0.0945127 | 2.866352 | 3.247013 |
| MIL | 154 | 18.20779 | 48.62067 | 0 | 277 |
| INV | 140 | 10.52195 | 1.983673 | 0 | 13.85976 |
| FCITY | 154 | 0.961039 | 1.282782 | 0 | 5 |
| DIS | 154 | 8.196776 | 0.2600556 | 7.753966 | 8.60853 |

The regression coefficient between the impact of material events and the perception of China's image is 0.0507, which passed the T-test. This indicates a positive correlation between the impact of material events and the perception of China's image. Specifically, the greater the difference between the number of material cooperation events and material conflict events, the more positive the perception of China's image by various countries. The impact of material events positively contributes to the perception of China's national image.

The regression coefficient between the impact of the scale of the South China Sea incidents and the perception of China's image is 0.2360, and it passed the T-test, indicating a significant positive correlation between the two. Specifically, the impact of the scale of the South China Sea incidents reflects the overall cooperative or conflict tendencies of the events, as well as their comprehensive influence on the regional situation. When the AVGgoldstein value is positive and higher, it indicates a greater intensity of cooperation in the South China Sea incidents, leading to a more positive perception of China among Southeast Asian countries. Conversely, when the AVGgoldstein value is negative and lower, it suggests a higher intensity of conflict, resulting in a more negative perception of China. Therefore, the impact of the scale of the South China Sea incidents positively contributes to the perception of China's national image.

The regression coefficients of control variables such as export scale and bilateral geographic distance with the perception of China's image are negative, and both passed the T-test, indicating that export scale and bilateral geographic distance play a suppressive role in shaping this perception. In contrast, the regression coefficients for the number of weapons imported from China, investment dependence, and the number of sister cities are not significant, suggesting that these factors have no significant impact on the perception of China's national image.

**3.2.3 Moderation effect regression.** Further moderation effect regression was conducted, and the results are shown in Table 4.

The results of the moderation effect regression show that each model passed the F-test at the 1% significance level, indicating a good overall regression performance.

The regression coefficient of verbal event impact on perceptions of China's image is 0.0042, and it is statistically significant according to the T-test. The regression coefficient of GDP on perceptions of China's image is -2.1374, and it also passes the T-test. Furthermore, the regression coefficient for the interaction term between verbal event impact and GDP is -0.0433, which is statistically significant. This indicates that GDP has a moderating effect on the relationship between verbal event impact and perceptions of China's image. Specifically, as the GDP of Southeast Asian countries increases, the positive influence of verbal event impact on their perception of China's image is weakened. A possible explanation is that with economic growth, Southeast Asian countries become more confident in international affairs. They are no longer content with mere diplomatic rhetoric or superficial gestures of friendship; instead, they seek tangible economic benefits

**Table 3. Benchmark regression results.**

| | AvgtoneY | AvgtoneY | AvgtoneY |
|---|---|---|---|
| VC | 0.0076*** | | |
| | (0.0019) | | |
| MC | | 0.0507*** | |
| | | (0.0185) | |
| AVGgoldstein | | | 0.2360** |
| | | | (0.1187) |
| EX | -81.9644*** | -72.9926*** | -71.4786*** |
| | (12.8297) | (13.2039) | (13.5351) |
| MIL | 0.0082 | 0.0094 | 0.0102 |
| | (0.0077) | (0.0080) | (0.0081) |
| INV | -0.2551 | -0.3592* | -0.3472* |
| | (0.1832) | (0.1880) | (0.1910) |
| FCITY | -0.0487 | 0.0330 | 0.0333 |
| | (0.1985) | (0.2052) | (0.2087) |
| DIS | -37.4584*** | -30.2496*** | -29.7585*** |
| | (5.4559) | (5.3778) | (5.4866) |
| _cons | 557.4635*** | 474.2296*** | 465.1205*** |
| | (81.0067) | (81.8667) | (83.7388) |
| $R^2$ | 0.3856 | 0.3415 | 0.3201 |
| F-statistics | 4.71*** | 3.89*** | 3.53*** |
| Hausman Test | 61.38 ( P=0.0000 ) | 39.95 ( P=0.0000 ) | 38.58 ( P=0.0000 ) |

Note: *, **, and *** denote significance levels at 10%, 5%, and 1%, respectively. Values in parentheses are robust standard errors.

and cooperation outcomes. As a result, their evaluation criteria for verbal events become more stringent, diminishing the effectiveness of verbal gestures in enhancing their perception of China's image.

The regression coefficient of the impact of material events on perceptions of China's image is 0.0535, which is statistically significant according to the T-test. The regression coefficient of GDP on perceptions of China's image is -1.2651, and it also passes the T-test. Additionally, the regression coefficient for the interaction term between the impact of material events and GDP is -0.6109, which is statistically significant. This indicates that GDP moderates the influence of material events on perceptions of China's image. Specifically, as the GDP of Southeast Asian countries increases, the positive impact of material events on their perception of China's image is weakened. One possible explanation is that, with the growth in GDP, Southeast Asian countries place greater emphasis on whether practical cooperation has directly contributed to their economic development and improved the living standards of their populations, rather than merely focusing on the signing of cooperation agreements. This pragmatic approach leads to stricter assessments of material events, thereby reducing the positive influence of these events on perceptions of China's image. Furthermore, as GDP increases, Southeast Asian countries also become more competitive in the international market. They no longer solely rely on China's investment and technological support; instead, they aim to achieve their development goals through independent growth and diversified international cooperation. This heightened sense of competition makes them more cautious about China's economic influence during practical cooperation, thereby diminishing the positive impact of material events on perceptions of China's image.

The regression coefficient for the impact of the South China Sea event scale on perceptions of China's image is 0.1383, which is statistically significant according to the T-test. The regression coefficient of GDP on perceptions of

**Table 4. Moderation effect regression.**

| | AvgtoneY | AvgtoneY | AvgtoneY |
|---|---|---|---|
| VC | 0.0042*** | | |
| | (0.0008) | | |
| MC | | 0.0535*** | |
| | | (0.0178) | |
| AVGgoldstein | | | 0.1383* |
| | | | (0.0793) |
| GDP | -2.1374*** | -1.2651*** | -1.5173*** |
| | (0.2371) | (0.3832) | (0.2872) |
| VC_GDP | -0.0433*** | | |
| | (0.0133) | | |
| MC*GDP | | -0.6109** | |
| | | (0.2967) | |
| AVGgoldstein*GDP | | | -3.0119** |
| | | | (1.3944) |
| EX | -7.6453 | -30.4454** | -19.9395* |
| | (12.0760) | (14.6213) | (10.7957) |
| MIL | 0.0063 | 0.0091** | 0.0069* |
| | (0.0050) | (0.0042) | (0.0040) |
| INV | -0.3074** | -0.4864** | -0.3710** |
| | (0.1319) | (0.1895) | (0.1525) |
| FCITY | -0.1073 | 0.1057 | 0.1389 |
| | (0.0931) | (0.1463) | (0.1413) |
| DIS | -62.2936*** | -42.5390*** | -42.8349*** |
| | (3.8526) | (5.1532) | (4.6296) |
| _cons | 1219.4295*** | 854.3959*** | 904.7994*** |
| | (82.8580) | (124.3865) | (104.6896) |
| $R^2$ | 0.8409 | 0.8645 | 0.8853 |
| F-statistics | 34.02*** | 18.35*** | 20.26*** |

Note: *, **, *** represent significance levels at 10%, 5%, and 1% respectively, with robust standard errors in parentheses

China's image is -1.5173, and it also passes the T-test. Additionally, the regression coefficient for the interaction term between the impact of the South China Sea event scale and GDP is -3.0119, which is statistically significant. This indicates that GDP moderates the effect of the South China Sea event scale on perceptions of China's image. Specifically, as the GDP of Southeast Asian countries increases, the positive influence of the South China Sea event scale on their perception of China's image is weakened. The underlying reasons for this phenomenon include the following: Firstly, as GDP rises, Southeast Asian countries become more autonomous and confident in geopolitical affairs. This is especially true for those with sovereignty disputes with China in the South China Sea. Their stance on the issue becomes more resolute and independent, and even if China demonstrates a high willingness to cooperate on South China Sea matters, these countries may adopt a more cautious approach, assessing whether such cooperation aligns with their long-term interests. Additionally, rapid economic growth increases Southeast Asian countries' focus on sovereignty and security issues. Even when China shows a high degree of cooperation, they are more concerned with how to safeguard their own sovereignty and security within such collaborations. As a result, their assessments of China's image tend to be more complex and cautious, thereby diminishing the positive impact of the South China Sea event scale on perceptions of China's image.

 

In conclusion, GDP, as a moderating variable, demonstrates a significant moderating effect in the relationship between speech events, concrete events, and the scale of South China Sea events on the perception of China's image. The increase in GDP among Southeast Asian countries leads them to adopt higher standards and a more pragmatic attitude when evaluating the impact of China's speech events, concrete events, and the scale of South China Sea events, thereby diminishing the positive influence of these events on China's image. The detailed analysis further reveals the roles of economic confidence, economic independence, pragmatic economic interests, competitive awareness, autonomous geopolitical strategies, and considerations of sovereignty and security in this process. These findings hold substantial theoretical and practical significance for understanding the complex relationship between Southeast Asian countries and China.

**3.2.4 Heterogeneity regression.** The study further divides the research sample into countries with territorial disputes with China over the South China Sea and those without such disputes, conducting a heterogeneity regression analysis (The countries that have sovereign claims and territorial disputes with China in the South China Sea include Vietnam, the Philippines, Malaysia, and Brunei. Additionally, due to the overlap of exclusive economic zones (EEZs) between Indonesia and China, this study also categorizes Indonesia as a country with territorial disputes with China). The heterogeneity results are presented in Table 5.

The analysis results of the variable of verbal event influence indicate that for countries with territorial disputes with China, the regression coefficient between verbal event influence and image perception is 0.0065, and it is significant. This means that for countries with territorial disputes with China, verbal event influence has a positive effect on their perception of China's image. For countries without territorial disputes with China, the regression coefficient between verbal event influence and image perception is 0.0780, and it is significant. This indicates that for countries without

**Table 5. Heterogeneity regression results.**

| AvgtoneY | Existence of Dispute | Absence of Dispute | Existence of Dispute | Absence of Dispute | Existence of Dispute | Absence of Dispute |
|---|---|---|---|---|---|---|
| VC | 0.0065*** | 0.0780*** | | | | |
| | (0.0019) | (0.0220) | | | | |
| MC | | | 0.0544*** | 0.0506 | | |
| | | | (0.0200) | (0.0459) | | |
| AVGgoldstein | | | | | 0.5134** | 0.2452*** |
| | | | | | (0.2266) | (0.0808) |
| EX | -73.1904*** | -68.8831*** | -58.6757*** | -91.8885*** | -62.8295*** | -91.2459*** |
| | (16.5331) | (20.9810) | (16.8410) | (21.7377) | (13.8704) | (15.6735) |
| MIL | 0.0229 | 0.0065 | 0.0311 | 0.0058 | 0.0381 | 0.0039 |
| | (0.0402) | (0.0076) | (0.0420) | (0.0083) | (0.0230) | (0.0061) |
| INV | -0.6430** | -0.0738 | -0.9201*** | -0.0615 | -0.9559*** | -0.0333 |
| | (0.3068) | (0.2158) | (0.3113) | (0.2360) | (0.2848) | (0.1866) |
| FCITY | -0.0781 | -0.0877 | 0.1020 | -0.0230 | -0.1035 | 0.0578 |
| | (0.2866) | (0.2618) | (0.2994) | (0.2857) | (0.1821) | (0.2037) |
| DIS | -36.3411*** | 0.6258 | -28.5101*** | 2.2555 | -32.1835*** | 2.2207 |
| | (6.6887) | (2.2401) | (6.6130) | (2.4875) | (5.5024) | (1.8628) |
| _cons | 525.0228*** | 215.3597*** | 420.8853*** | 276.1142*** | 463.4701*** | 273.5795*** |
| | (101.8379) | (70.5287) | (102.3457) | (74.4843) | (85.0463) | (55.5276) |
| R2 | 0.4964 | 0.4188 | 0.4559 | 0.3055 | 0.7699 | 0.6271 |
| F-statistics | 5.30*** | 4.11*** | 4.50*** | 2.51** | 17.99*** | 9.59*** |

Note: ***, **, * denote significance levels at 1%, 5%, and 10%, respectively, with robust standard errors in parentheses.

sovereignty disputes or territorial conflicts with China, verbal event influence has a positive effect on their image perception. Moreover, for countries without territorial disputes with China, the positive impact of verbal event influence on their perception of China's image is more pronounced. This may be because Southeast Asian countries without sovereignty disputes with China in the South China Sea tend to use proactive diplomatic measures and positive media coverage to enhance their international image, which makes the positive effect of verbal events on their perception of China's image more significant [28]. Additionally, countries without South China Sea disputes with China might be more sensitive to verbal events involving diplomacy and international conflict, leading them to choose their words more carefully when dealing with international incidents, thereby strengthening the positive influence of verbal events on their perception of China's image. Furthermore, domestic public opinion and the political environment play an important role in shaping diplomatic policies and international image. Countries without sovereignty disputes with China in the South China Sea may face less external pressure and negative public opinion domestically, making it easier for them to use proactive diplomatic and verbal strategies to promote the positive development of their international image [29]. Huang, Cook, and Xie (2024) explored the mediating role of the media in public opinion toward China, while Mattingly and Sundquist (2023) investigated the effectiveness of public diplomacy in the context of "wolf warrior diplomacy." Different from these studies, this research focuses on the impact of the South China Sea issue on Southeast Asian countries' perception of China's image. It specifically targets Southeast Asian countries and uses GDELT big data and a panel multiple linear regression model for analysis from a unique perspective and method, aiming to fill the gaps in relevant research.

The analysis of the variable representing the impact of material events indicates that for countries with territorial disputes with China, the regression coefficient between real event impact and image perception is 0.0544, and it is significant. This suggests that for countries with sovereignty disputes with China, real event impact positively influences their perception of China's image. However, for countries without territorial disputes with China, the regression coefficient between real event impact and image perception is 0.0506, but it is not significant. This implies that for countries without territorial disputes with China, the influence of material events on their perception of China's image is not evident. The possible reason for this is that Southeast Asian countries with territorial and sovereignty disputes with China have higher political and security sensitivities and pay closer attention to material events involving China in the South China Sea. Related media are more inclined to closely follow and report on these events, often with strong emotional tones and political biases, which can significantly shape public perceptions and sentiments regarding these events.

The results of the heterogeneity analysis reveal that the impact of the South China Sea event scale on Southeast Asian countries' perceptions of China's image varies. For countries with territorial disputes with China, the regression coefficient between the South China Sea event scale and their perception of China's image is 0.5134, and it is significant, indicating that the scale of South China Sea events has a significantly positive impact on these countries' perceptions of China's image. In contrast, for countries without territorial disputes with China, the regression coefficient is 0.2452, which is also significant. This suggests that even in the absence of territorial disputes, the scale of South China Sea events can improve these countries' perceptions of China's image. However, the positive impact is more pronounced for countries with territorial disputes. A possible explanation is that these countries are more attentive and sensitive to events in the South China Sea, making them more inclined to engage with China in dispute resolution. Therefore, an increase in the scale of South China Sea events is more likely to improve these countries' perceptions of China's image.

**3.2.5 Robustness test.** The study excludes the research samples from 2020 to 2022 to eliminate the impact of the pandemic and conducts a robustness regression. The results of the robustness regression are shown in Table 6.

The results of the robustness regression indicate that the regression coefficients of the impact of verbal events, the impact of material events, and the impact of the scale of South China Sea events on the perception of China's image are all positive, consistent with the previous regression results. This suggests that the regression results of this study are robust.

**Table 6. Robustness regression results.**

| | AvgtoneY | AvgtoneY | AvgtoneY |
|---|---|---|---|
| VC | 0.0070*** | | |
| | (0.0022) | | |
| MC | | 0.0295*** | |
| | | (0.0087) | |
| AVGgoldstein | | | 0.2068** |
| | | | (0.0813) |
| EX | -117.6117*** | -111.3563*** | -119.7755*** |
| | (17.9293) | (11.0883) | (13.0779) |
| MIL | 0.0008 | -0.0013 | -0.0032 |
| | (0.0105) | (0.0069) | (0.0065) |
| INV | -0.2810 | -0.4664** | -0.4765** |
| | (0.2026) | (0.2049) | (0.2147) |
| FCITY | -0.0093 | -0.0079 | -0.0664 |
| | (0.2265) | (0.1534) | (0.1695) |
| DIS | -47.9571*** | -42.9577*** | -45.8570*** |
| | (6.9888) | (4.1704) | (4.7764) |
| _cons | 753.1912*** | 697.0548*** | 746.5430*** |
| | (109.1034) | (64.9222) | (76.5074) |
| $R^2$ | 0.4302 | 0.7789 | 0.6713 |
| F-statistics | 4.15*** | 19.38*** | 11.23*** |

Note: ***, **, * denote significance levels at 1%, 5%, and 10%, respectively, with robust standard errors in parentheses.

## 4 Countermeasures and recommendations

How does the "South China Sea issue" influence China's image in Southeast Asia, and does it diminish Southeast Asian countries' perceptions of China? Based on the sentiment index data of Southeast Asian countries toward China from the Global Database of Events, Language, and Tone (GDELT) from 2010 to 2024, this study employs a panel linear multiple regression model to empirically examine the impact of the "South China Sea issue" on Southeast Asian countries' perceptions of China's image. After controlling for bilateral factors such as economic, military, and diplomatic ties, the study finds that: (1) The "South China Sea issue" significantly affects Southeast Asian countries' perceptions of China's image, with verbal events, real-world events, and the scale of South China Sea events all exerting a substantial influence on China's overall image; (2) Although economic cooperation can partially mitigate negative perceptions, unresolved sovereignty disputes and territorial conflicts in the context of the "South China Sea issue" continue to significantly deteriorate Southeast Asian countries' overall impressions of China. GDP also plays a significant moderating role in this process; as Southeast Asian countries' economic power increases, the positive impact of the South China Sea issue on their perceptions of China's image gradually diminishes. (3) There are significant differences in how the "South China Sea issue" affects perceptions of China's image among different Southeast Asian countries, with its impact being more pronounced in countries with territorial disputes with China. Based on these findings, this study offers the following recommendations.

(1) **Strengthening Cultural Exchanges to Enhance Verbal Cooperation.** The unresolved "South China Sea issue" has resulted in generally negative perceptions of China among Southeast Asian countries, especially those with territorial disputes and sovereignty conflicts with China. China should enhance its public diplomacy, promoting the concept of peaceful development and clarifying misunderstandings. At the same time, it is essential to strengthen cultural

exchanges with Southeast Asian countries, deepen interactions among youth, and foster people-to-people connections through educational programs, cultural exchange activities, and personnel exchanges, thereby enhancing cultural affinity and emotional ties between Southeast Asian countries and China. Furthermore, China should engage in positive verbal interactions with Southeast Asian countries, including conveying positive messages during international conferences and bilateral talks, and using media to promote China's friendly stance and willingness to cooperate. Systematically promoting China's vision of peaceful development and intentions can help dispel misunderstandings and biases toward China. In particular, when engaging with countries involved in South China Sea disputes with China, it is crucial to utilize diverse media channels to comprehensively showcase China's positive image, convey positive messages, and strive to reduce negative perceptions caused by information asymmetry.

(2) **Promote Pragmatic Cooperation and Strengthen Practical Interests.** Firstly, focus on tangible outcomes to improve the well-being of the people. In collaboration with Southeast Asian countries, special attention should be given to the practical impact and sustainability of projects. By engaging in pragmatic cooperation in areas such as education, healthcare, infrastructure, and digital technology, China can improve the living standards of local populations, thereby enhancing their goodwill and trust towards China. This is especially crucial when working with countries like the Philippines and Vietnam, which have sovereignty disputes and territorial conflicts with China. The success of practical projects can help alleviate hostile sentiments and improve negative perceptions. Secondly, enhance security cooperation and establish strategic trust mechanisms. Further promote security cooperation with Southeast Asian countries, particularly in maritime security, counter-terrorism, and anti-piracy efforts. Establish regular communication mechanisms, and undertake joint maritime patrols and rescue operations to bolster regional security and stability, presenting China as a responsible major power and easing tensions caused by the South China Sea issues. Finally, advance economic cooperation and connectivity, further deepening economic ties with Southeast Asian countries through initiatives like the Belt and Road Initiative (BRI) and the Regional Comprehensive Economic Partnership (RCEP). Strengthening economic bonds can foster political trust through economic interests, thereby mitigating the negative impact of South China Sea issues on China's image.

(3) **Flexibly Respond to Diplomatic Challenges and Innovate New Cooperation Models**. Firstly, adopt strategies tailored to local conditions and adjust cooperation strategies flexibly. Based on the varying levels of economic development and dependency on the Chinese market among Southeast Asian countries, adjust cooperation strategies accordingly. For countries with higher economic autonomy, adopt a win-win cooperation model to promote mutual development. For those with higher economic dependency, provide increased technical and developmental support to help them strengthen their economic independence, thereby reducing their wariness of China's economic influence. Secondly, innovate cooperation forms and explore diversified platforms. Consider establishing new cooperation mechanisms, such as regional development funds or joint development zones, to provide more diversified cooperation platforms. This could promote joint development and resource sharing in the South China Sea dispute areas, reduce negative perceptions of China resulting from singular cooperation models, and foster regional economic integration and shared prosperity.

(4) **Strengthen Crisis Management and Emergency Response Mechanisms.** Firstly, promote the establishment of a regional dispute resolution mechanism, aiming to resolve disputes through legal and diplomatic channels, thus preventing the escalation of real-world conflicts and ensuring that disputes are resolved within a peaceful framework. Secondly, establish a regional crisis management center to enhance the ability to respond to emergencies and improve coordination through joint training and exercises. This ensures that in the event of sudden real-world incidents, all parties can respond swiftly, minimizing losses and reducing the risk of conflict escalation.

(5) **Strengthen Institutional Constraints and Optimize Dispute Resolution.** Enhance the construction and improvement of regional multilateral cooperation mechanisms, actively implement the *Declaration on the Conduct of Parties*

*in the South China Sea* (DOC) and promote the adoption of the *Code of Conduct in the South China Sea* (COC) as well as international rules and standards. This involves establishing and strengthening cooperation platforms between ASEAN and China, formulating clear conduct norms and dispute resolution procedures. Additionally, encourage the international community to promote the internationalization of norms and standards concerning the South China Sea, ensuring that dispute resolution aligns with international rules and standards. This approach can help improve China's credibility and image in the international community, thereby mitigating the negative perceptions of Southeast Asian countries toward China.

Based on the significant role of the "South China Sea issue" in shaping China's national image, especially verbal events, material events, and the scale of South China Sea incidents will all have a profound impact on the perceptions of Southeast Asian countries to varying degrees. China needs to manage the differences and conflicts arising from the "South China Sea issue" while reframing China-Southeast Asia relations and deconstructing the strategic narrative between China and Southeast Asia. This can be achieved by continuing to promote initiatives such as the "Community of Shared Future for Mankind," "Global Partnership," "Global Development Initiative," "Global Security Initiative," and "Chinese-style Modernization." These efforts aim to deconstruct, replace, and counter the negative narratives and framing of China's national image that are shaped by theories such as the "Thucydides Trap," "Power Transition Theory," "Great Power Hegemony," and the "Clash of Civilizations." While the unresolved nature of the "South China Sea issue" means that cooperative interactions between China and Southeast Asian countries cannot directly eliminate perceptions of threat and the "hostility spiral" toward China, verbal cooperation, real cooperation, and the influence of South China Sea events can mitigate Southeast Asian countries' misconceptions about the "China threat." A cooperative narrative regarding the "South China Sea issue" and the strategic use of metaphorical substitution, discourse construction, and substantive collaboration to counter conflict-based narratives can help reduce the emotional opposition and negative inferences about China's national image among some Southeast Asian countries that view China as a "rising power." China should adopt a multi-layered and multi-dimensional approach to cooperation and innovative narratives to gradually mitigate the negative impact of the "South China Sea issue" on its national image. By advancing pragmatic cooperation, deepening cultural exchanges, and innovating diplomatic models, China can build a more solid foundation of regional trust, thereby fostering long-term stability and sustainable development in relations with Southeast Asian countries. This strategy not only effectively manages differences and conflicts but also injects new momentum into regional peace and prosperity, ultimately enhancing China's international image and strategic standing.

## 5 Discussion

In conclusion, this study underscores the complex interplay between geopolitical issues, economic development, and national image perception. By shedding light on these dynamics, it aims to facilitate more nuanced and effective diplomatic strategies and to promote stable and sustainable development of China-Southeast Asia relations. Future research could further explore the long-term effects of these dynamics and the potential for collaborative initiatives to mitigate negative perceptions and enhance mutual understanding.

Although this study, based on GDELT big data, reveals the significant impact of the South China Sea issue on Southeast Asian countries' perceptions of China's image, there are still some shortcomings and limitations. First, the GDELT data relies on media reports, which may be affected by media tendency and information selectivity, and the uneven coverage in different countries leads to the problem of data bias and insufficient sample size. Second, in terms of variable selection, although control variables such as economy, military and diplomacy are included, non-economic and non-political factors such as cultural exchanges and civil interactions are not fully taken into account, and some variables such as GDP may not fully reflect the country's economic structure and development level. In addition, it is difficult to fully determine the direction of causality in the model setting, and the heterogeneity analysis fails to fully take into account the complex differences among Southeast Asian countries in terms of political system, ethnic composition and historical

background. In terms of research perspectives, the study mainly focuses on international relations and geopolitics, ignoring the micro-mechanisms and psychological factors of other perspectives such as social psychology and cultural anthropology, and the study is limited to Southeast Asia and lacks comparative analysis with other regions. In terms of time span, the study fails to adequately distinguish between the short-term and long-term impacts of the South China Sea issue at different stages. Future research can expand and deepen the study in terms of diversification of data sources, improvement of the variable system, in-depth exploration of causality, fusion of multidisciplinary perspectives, regional comparative analysis, and differentiation between long-term and short-term impacts in order to more comprehensively and accurately reveal the mechanism of impacts and dynamics of the South China Sea Issue on the perceptions of Southeast Asian countries.

## 6 Conclusion

This study has comprehensively investigated the impact of the South China Sea issue on Southeast Asian countries' perception of China's image using a panel multiple linear regression model based on GDELT big data from 2010 to 2024. The key findings and contributions of this research can be summarized as follows:

Significant Impact of the South China Sea Issue: The empirical results demonstrate a substantial positive correlation between the South China Sea issue and Southeast Asian countries' perception of China's image. Verbal events, material events, and the scale of events related to the South China Sea significantly enhance the positive evaluation and recognition of China among these countries. This indicates that the South China Sea issue plays a crucial role in shaping China's national image in the region.

Moderating Role of GDP: GDP acts as a moderating variable in this relationship. As the GDP of Southeast Asian countries increases, the positive impact of verbal events, material events, and the scale of South China Sea events on their perception of China's image diminishes. This suggests that economic development influences how these countries perceive and evaluate China's actions and intentions.

Differences Among Southeast Asian Countries: The study reveals notable differences in the perception of China's image among Southeast Asian countries. Nations with territorial disputes and sovereignty conflicts with China, such as Vietnam and the Philippines, exhibit more pronounced negative perceptions, while countries like Laos, Cambodia, and Singapore have more positive perceptions. This variation is influenced by geopolitical, historical, and economic factors unique to each country.

Academic and Practical Contributions: This research contributes to the academic understanding of how geopolitical issues like the South China Sea dispute affect national image perception. It provides a foundation for future studies and offers valuable insights for policymakers in China and Southeast Asian countries. The findings highlight the importance of considering economic development levels and historical contexts when formulating policies aimed at improving bilateral relations.

## Author contributions

**Writing – original draft:** Zibo Wei, Xuan Chen, Genli Tang, Yishuai Xie.

**Writing – review & editing:** Zibo Wei, Xuan Chen, Genli Tang, Yishuai Xie.

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
