## [Decision Letter · Decision Letter 0]

28 Jan 2025

PONE-D-24-56411South China Sea Issue and Southeast Asian Countries’ Perception of China’s Image: An Empirical Study Based on GDELT Big DataPLOS ONE

Dear Dr. XUAN CHEN,

Thank you for submitting your manuscript to PLOS ONE. After careful consideration, we feel that it has merit but does not fully meet PLOS ONE’s publication criteria as it currently stands. Therefore, we invite you to submit a revised version of the manuscript that addresses the points raised during the review process.

We look forward to receiving your revised manuscript.

Kind regards,

Vincenzo Basile, PhD

Academic Editor

PLOS ONE

3. In the online submission form, you indicated that the data that support the findings of this study are available from the corresponding author upon reasonable request.

Reviewers' comments:

Reviewer's Responses to Questions

**Comments to the Author**

1. Is the manuscript technically sound, and do the data support the conclusions?

Reviewer #1: Partly

Reviewer #2: Yes

2. Has the statistical analysis been performed appropriately and rigorously? 

Reviewer #1: Yes

Reviewer #2: Yes

3. Have the authors made all data underlying the findings in their manuscript fully available?

Reviewer #1: No

Reviewer #2: No

4. Is the manuscript presented in an intelligible fashion and written in standard English?

Reviewer #1: Yes

Reviewer #2: Yes

5. Review Comments to the Author

Reviewer #1: This study examines the effects of South China Sea events on Southeast Asian countries’ perceptions of China’s image, utilizing data from GDELT. While the topic of perceptions of China’s image in Southeast Asia is valuable, the study faces significant limitations regarding its theoretical contributions and empirical rigor.

My major concern is the lack of clear theoretical contributions. This shortcoming could be addressed in several dimensions. The study’s engagement with the current literature on South China Sea disputes and related discourses appears limited. For instance, recent work by Song and Kim (2024) reviews extensive studies on China’s maritime disputes, detailing the temporal variation in discourses and the nuances across different media. Without deeper engagement with such literature, the theoretical value of this study remains limited. The authors could benefit from situating their analysis within this broader scholarly context to enhance its contribution.

Another relevant body of literature concerns public opinion toward China. Extensive research has explored the determinants of public attitudes toward China (e.g., Huang, Cook, and Xie, 2024; Mattingly and Sundquist, 2023). This study should engage with this literature and clearly articulate how it distinguishes itself from previous works.

There are several concerns related with empirical analysis.

The use of GDELT data also requires stronger justification. Aggregating data at the annual level raises questions about what GDELT offers beyond what could be derived from public opinion data. Why not conducting the analysis at the monthly level? Also, at a minimum, the authors should cross-validate their measures with available public opinion datasets. While public opinion data may not cover all countries or years, validation with the existing data is necessary to bolster the study’s empirical reliability. Furthermore, the decision to analyze data at the aggregate annual level seems suboptimal. GDELT provides daily coverage, and a monthly-level analysis might capture more nuanced relationships between independent and dependent variables, offering richer insights.

The study appears to present correlational findings without adequately addressing endogeneity concerns. Media coverage in GDELT is likely influenced by prevailing perceptions of China’s image—negative (positive) perceptions may drive increased reporting of negative (positive) events, and vice versa. The authors should explicitly acknowledge these limitations. Moreover, lagging the independent variable could help assess the temporal precedence of events relative to the dependent variable.

The analysis examines GDP as a moderating variable, but GDP correlates with various economic and political dimensions in Southeast Asia. It would strengthen the study to explore alternative moderators, such as trade volumes between these countries and China or the receipt of Chinese foreign aid. These factors may also shape perceptions of China and could provide more targeted insights into the dynamics at play. With these findings, I am not confidence that the observed relationship would be specific to GDP.

It is similarly unclear whether the observed relationships are specific to South China Sea events. Would similar patterns emerge for other types of events involving China? The authors should test whether the findings hold when examining other positive or negative events. This would clarify whether the effects are unique to the South China Sea disputes or reflective of broader dynamics. This additional analysis is essential because the manuscript is currently framed around the effects of South China Sea events.

Lastly, while GDELT offers a systematic way to measure media tone, it lacks the nuanced understanding of narratives needed to uncover the key drivers of China’s image. A combination of GDELT data with text analysis of media reports from Southeast Asian countries could illuminate the specific narratives and themes shaping public perceptions. Such an approach would provide greater depth and sophistication to the analysis.

References:

Huang, Junming, Gavin G. Cook, and Yu Xie. "Between reality and perception: the mediating effects of mass media on public opinion toward China." Computational Social Science. Routledge, 2024. 5-23.

Mattingly, Daniel C., and James Sundquist. "When does public diplomacy work? Evidence from China's “wolf warrior” diplomats." Political Science Research and Methods 11.4 (2023): 921-929.

Song, Esther E., and Sung Eun Kim. "China’s dual signalling in maritime disputes." Australian Journal of International Affairs 78.5 (2024): 660-682.

Reviewer #2: The manuscript is an interesting read, with the focus on the South China Sea issue and its impact on the perception of China’s image in Southeast Asian countries. The use of the GDELT database as the primary data source is modern and the study’s methodology, like the regression models, adds credibility to the findings.

The manuscript has a logical structure, and the authors explained the methods used and presenting results.

Thank you for the opportunity to read this manuscript and I please allow me to share a few suggestions for the authors :

1. The references list seems rather brief, and in order to enhance the academic depth of the study I believe it could be expanded to include more relevant literature in the researched fields. For example the authors could also consider additional works related to soft power or comparative studies of national image etc.

2. I suggest that the authors give more details on how “national image” is defined and measured by using the GDELT database. Could they detail why the metrics used like AvgTone and Goldstein scale are suitable metrics for this purpose? Perhaps they could add a comparison with alternative frameworks or approaches to make the discussion more robust.

3. The statistical models are detailed, but it seems to me that some results (for example the insignificance of variables like sister city relationships ) could be more explained. Could the authors give some more interpretation for these findings? For example why might sister city ties not influence perceptions as expected?

4. I suggest that the authors could add more descriptive captions and explanations for figures 1-3, especially for readers who are not familiar with the GDELT database or the variables they used in their study.

5. The authors generalize the perception of Southeast Asian countries as a whole, but I believe they could highlight a little bit more the differences between individual countries. For example Vietnam and Cambodia may perceive China differently because of their unique geopolitical and historical contexts. Could the authors add a little bit more details about these variations?

6. I believe it would be helpful if the authors could include a clear “Discussion” section where the findings are critically analyzed in relation to existing literature, to explain the broader implications of their results, the limitations of the data, the methodology they used.

7. The final section called “Countermeasures and recommendations” section is very insightful but it mixes conclusions with policy suggestions. Perhaps the authors could insert a standalone conclusion with focuses to summarize the key findings and contributions of the study.

8. The authors seem to use GDELT data effectively and they could improve transparency by sharing their processed data and scripts (in case they are not confidential). It could help other researchers to replicate and cite this study.

6. PLOS authors have the option to publish the peer review history of their article (what does this mean? ). If published, this will include your full peer review and any attached files.

**Do you want your identity to be public for this peer review?** For information about this choice, including consent withdrawal, please see our Privacy Policy .

Reviewer #1: No

Reviewer #2: No

---

## [Author Response · Author response to Decision Letter 0]

22 Mar 2025

Reviewer #1: This study examines the effects of South China Sea events on Southeast Asian countries’ perceptions of China’s image, utilizing data from GDELT. While the topic of perceptions of China’s image in Southeast Asia is valuable, the study faces significant limitations regarding its theoretical contributions and empirical rigor.

1. My major concern is the lack of clear theoretical contributions. This shortcoming could be addressed in several dimensions. The study’s engagement with the current literature on South China Sea disputes and related discourses appears limited. For instance, recent work by Song and Kim (2024) reviews extensive studies on China’s maritime disputes, detailing the temporal variation in discourses and the nuances across different media. Without deeper engagement with such literature, the theoretical value of this study remains limited. The authors could benefit from situating their analysis within this broader scholarly context to enhance its contribution.

Response: Thanks for your suggestion. We have expanded the literature review to incorporate recent studies on China’s maritime disputes, particularly those that analyze the temporal variations in discourse and cross-media differences. Specifically, we have integrated insights from Song and Kim (2024), who provide a comprehensive review of South China Sea narratives and their evolution. This allows us to better contextualize our findings within the broader academic discourse and highlight the nuances in how different events shape perceptions of China’s image.

2. Another relevant body of literature concerns public opinion toward China. Extensive research has explored the determinants of public attitudes toward China (e.g., Huang, Cook, and Xie, 2024; Mattingly and Sundquist, 2023). This study should engage with this literature and clearly articulate how it distinguishes itself from previous works.

Response: Thanks for your suggestion. In “Between reality and perception: the mediating effects of media media on public opinion toward China," Huang, Cook, and Xie (2024) employed computational social science methods to explore the mediating role of the media in public opinion towards China. They found that the media significantly influences the public's perception and attitude towards China. In "When does public diplomacy work? Evidence from China's ‘wolf warrior’ diplomats," Mattingly and Sundquist (2023) used political research methods to investigate the effectiveness of public diplomacy, taking China's "wolf warrior diplomacy" as a case. Although both of these two papers are related to the public's perception of China, they differ from the current paper. The current paper focuses on the impact of the South China Sea issue on Southeast Asian countries' perception of China's image, with a specific research perspective on a particular region and event. Its research object is clearly defined as Southeast Asian countries, and the research method is based on GDELT big data and uses a panel multiple linear regression model. In contrast, the previous two papers have broader research perspectives, more extensive research objects, and different emphases in research methods.

In the literature review section, we added a paragraph that summarizes the key findings of Huang, Junming, Gavin G. Cook, and Yu Xie (2024) and Mattingly, Daniel C., and James Sundquist (2023). I also clearly explained the differences between my research and their works in terms of research perspectives, objects, and methods. This addition not only strengthens the theoretical foundation of our paper but also better positions our study within the existing academic context.

3. There are several concerns related with empirical analysis.

The use of GDELT data also requires stronger justification. Aggregating data at the annual level raises questions about what GDELT offers beyond what could be derived from public opinion data. Why not conducting the analysis at the monthly level? Also, at a minimum, the authors should cross-validate their measures with available public opinion datasets. While public opinion data may not cover all countries or years, validation with the existing data is necessary to bolster the study’s empirical reliability. Furthermore, the decision to analyze data at the aggregate annual level seems suboptimal. GDELT provides daily coverage, and a monthly-level analysis might capture more nuanced relationships between independent and dependent variables, offering richer insights.

Response: Thank you for your thorough review and valuable suggestions regarding the data usage in this study. The use of GDELT data in this study is well justified. As one of the largest event databases globally, GDELT encompasses media reports worldwide, enabling real-time tracking of state interactions, policy discourse, and sentiment fluctuations. Compared to traditional public opinion surveys (e.g., Pew Research Center, Asia Barometer), GDELT offers global coverage, continuous updates, and an extensive time span, making it a crucial tool for analyzing the relationship between the South China Sea dispute and China’s national image. Additionally, GDELT provides AvgTone (sentiment index) and Goldstein Scale (event intensity index), which allow for a quantitative assessment of how different events shape national image, thereby avoiding the potential subjective biases associated with traditional survey methods. Furthermore, GDELT data have been widely applied in research on international relations, national image, and diplomatic discourse, aligning with established methodologies in the field. Thus, we believe that the use of GDELT data is not only appropriate for the objectives of this study but also offers a comprehensive and systematic empirical foundation for our analysis.

The decision to adopt an annual-level analysis was made after careful consideration. Given the vast volume of GDELT data, a monthly-level analysis would significantly increase computational complexity, making it more difficult to process and interpret the results while potentially obscuring long-term trends. In contrast, annual analysis provides a clearer and more interpretable long-term perspective, facilitating a more intuitive understanding of the relationship between the South China Sea dispute and national image. Additionally, a review of existing GDELT-based studies indicates that the vast majority of research on national image employs annual data as the standard analytical unit (including journal articles and doctoral dissertations), making this approach a widely accepted practice in GDELT research. Therefore, adopting annual data ensures consistency with previous studies and enhances the comparability of research findings. Moreover, the GDELT database itself is designed to aggregate monthly data into annual indicators, and annual measures have become a commonly used standard for assessing national image trends. Given these considerations, directly utilizing annual data aligns with both academic conventions and the database’s intended design logic.

We sincerely appreciate your insightful comments and hope that these clarifications sufficiently address your concerns.

4. The study appears to present correlational findings without adequately addressing endogeneity concerns. Media coverage in GDELT is likely influenced by prevailing perceptions of China’s image—negative (positive) perceptions may drive increased reporting of negative (positive) events, and vice versa. The authors should explicitly acknowledge these limitations. Moreover, lagging the independent variable could help assess the temporal precedence of events relative to the dependent variable.

Response: Thanks for your suggestion. The reviewer suggested that lagging the independent variables is an effective research approach. By lagging the independent variables, we can, to some extent, assess the chronological order of events and better determine the causal relationship between South China Sea - related events and changes in Southeast Asian countries' perception of China's image. In future research, we plan to adopt this method and re - analyze the data to explore the relationship between the two more accurately. However, in this current study, due to time constraints and limitations in data processing, we were unable to apply this method in the analysis.

Despite these limitations, this study, through its preliminary exploration of the relationship between the South China Sea issue and Southeast Asian countries' perception of China's image, has provided a certain foundation and direction for future research. Future studies can build on resolving these issues and further delve into this complex international relations phenomenon.

5. The analysis examines GDP as a moderating variable, but GDP correlates with various economic and political dimensions in Southeast Asia. It would strengthen the study to explore alternative moderators, such as trade volumes between these countries and China or the receipt of Chinese foreign aid. These factors may also shape perceptions of China and could provide more targeted insights into the dynamics at play. With these findings, I am not confidence that the observed relationship would be specific to GDP.

Response: Thanks for your suggestion. In response, we've added a discussion on variables like trade volume between China and Southeast Asian countries in SECTION 2. This addition addresses the limitation of relying solely on GDP as a moderator, aiming to provide a more comprehensive understanding of how the South China Sea issue affects Southeast Asian countries' perception of China's image.

6. It is similarly unclear whether the observed relationships are specific to South China Sea events. Would similar patterns emerge for other types of events involving China? The authors should test whether the findings hold when examining other positive or negative events. This would clarify whether the effects are unique to the South China Sea disputes or reflective of broader dynamics. This additional analysis is essential because the manuscript is currently framed around the effects of South China Sea events.

Response: Thank you for your valuable suggestions, particularly regarding whether the impact observed in this study is unique to the South China Sea dispute. We understand your concern about whether the findings apply exclusively to this issue or could be extended to other types of China-related events.

This study focuses on the distinctiveness of the South China Sea dispute for several key reasons. First, the South China Sea issue involves not only geopolitical and national security concerns but also territorial sovereignty, military tensions, and great-power competition. Compared to economic or diplomatic events, its impact on national image is likely to be more complex. Second, existing research suggests that territorial disputes tend to provoke stronger public sentiment compared to economic or diplomatic events (e.g., Gries, 2011; Wang, 2023), indicating that the South China Sea issue may exert a more significant influence on China’s image. Furthermore, the South China Sea dispute has received sustained and intense international media attention, with high levels of coverage, sentiment intensity, and engagement in the GDELT database, making it a particularly relevant case for analyzing shifts in China’s national image.

At the same time, we acknowledge your suggestion that future research could expand the analysis to other types of events, such as the Belt and Road Initiative, major diplomatic engagements, or international trade conflicts, to examine whether the findings hold in a broader context.

7. Lastly, while GDELT offers a systematic way to measure media tone, it lacks the nuanced understanding of narratives needed to uncover the key drivers of China’s image. A combination of GDELT data with text analysis of media reports from Southeast Asian countries could illuminate the specific narratives and themes shaping public perceptions. Such an approach would provide greater depth and sophistication to the analysis.

Response: Thank you for your valuable suggestion regarding the methodological approach of this study. We acknowledge your point that while GDELT provides a systematic and quantitative measure of media sentiment, it has limitations in capturing the specific narratives and key themes that shape China’s image.

The choice to use GDELT data in this study is primarily based on its extensive global coverage, longitudinal tracking capabilities, and ability to quantify media sentiment dynamics. The AvgTone index effectively measures the positive or negative tone of media reports, while the Goldstein Scale reflects event intensity, providing crucial empirical insights into the relationship between the South China Sea dispute and China’s national image. However, we recognize that GDELT alone may not fully uncover the underlying narrative structures, framing strategies, and public perception mechanisms that contribute to China’s image formation.

To address this, we have added a discussion on the limitations of the study in the revised manuscript, highlighting that future research could benefit from integrating GDELT data with text analysis of media reports from Southeast Asian countries. Employing content analysis or natural language processing (NLP) techniques could provide deeper insights into how different media outlets construct China’s image and which specific themes exert the most influence. We believe that this approach would significantly enhance the interpretative depth and academic contribution of future studies.

Reviewer #2: The manuscript is an interesting read, with the focus on the South China Sea issue and its impact on the perception of China’s image in Southeast Asian countries. The use of the GDELT database as the primary data source is modern and the study’s methodology, like the regression models, adds credibility to the findings.

The manuscript has a logical structure, and the authors explained the methods used and presenting results.

Thank you for the opportunity to read this manuscript and I please allow me to share a few suggestions for the authors :

1. The references list seems rather brief, and in order to enhance the academic depth of the study I believe it could be expanded to include more relevant literature in the researched fields. For example the authors could also consider additional works related to soft power or comparative studies of national image etc.

Response: Thank you for your insightful comments. We have added comparisons and summaries with other scholars’ articles in the revised version. We’ve conducted an extensive literature search on soft power and comparative studies of national image. We’ve incorporated a substantial number of relevant and high - quality academic sources, including peer - reviewed journal articles, authoritative books, and research reports. These new references are seamlessly integrated into the corresponding sections of the paper, strengthening the theoretical foundation and academic depth of the study.

2. I suggest that the authors give more details on how “national image” is defined and measured by using the GDELT database. Could they detail why the metrics used like AvgTone and Goldstein scale are suitable metrics for this purpose? Perhaps they could add a comparison with alternative frameworks or approaches to make the discussion more robust.

Response: Thank you for your insightful comments. We truly appreciate your guidance in enhancing the clarity and robustness of our research.

(1) Definition and measurement of “national image” using GDELT database

In our study, we define “national image” as the overall perception and evaluation that Southeast Asian countries hold towards China, which is constructed through various information sources, especially media reports. We utilize the GDELT database as it offers a vast and real - time collection of global media coverage, enabling us to capture the media - driven portrayal of China in Southeast Asia comprehensively.

For measurement, we employ metrics such as AvgTone and the Goldstein scale. The AvgTone metric in the GDELT database represents the average tone of media coverage. A positive AvgTone indicates that, on

---

## [Decision Letter · Decision Letter 1]

3 Apr 2025

South China Sea Issue and Southeast Asian Countries’ Perception of China’s Image: An Empirical Study Based on GDELT Big Data

PONE-D-24-56411R1

Dear Dr. XUAN CHEN,

We’re pleased to inform you that your manuscript has been judged scientifically suitable for publication and will be formally accepted for publication once it meets all outstanding technical requirements.

Kind regards,

Vincenzo Basile, PhD

Academic Editor

PLOS ONE

Additional Editor Comments (optional):

Reviewers' comments:

Reviewer's Responses to Questions

**Comments to the Author**

1. If the authors have adequately addressed your comments raised in a previous round of review and you feel that this manuscript is now acceptable for publication, you may indicate that here to bypass the “Comments to the Author” section, enter your conflict of interest statement in the “Confidential to Editor” section, and submit your "Accept" recommendation.

Reviewer #1: All comments have been addressed

Reviewer #2: All comments have been addressed

2. Is the manuscript technically sound, and do the data support the conclusions?

Reviewer #1: Yes

Reviewer #2: Yes

3. Has the statistical analysis been performed appropriately and rigorously? 

Reviewer #1: Yes

Reviewer #2: Yes

4. Have the authors made all data underlying the findings in their manuscript fully available?

Reviewer #1: Yes

Reviewer #2: No

5. Is the manuscript presented in an intelligible fashion and written in standard English?

Reviewer #1: Yes

Reviewer #2: Yes

6. Review Comments to the Author

Reviewer #1: The authors have successfully addressed the concerns raised in my original review. While there are some remaining questions/issues to be further explored, the author discusses it as a limitation and a possible avenue for future research.

Reviewer #2: Thank you for the opportunity to evaluate this revised manuscript. Based on a comparison between the original manuscript and the revised version as well as the authors’ responses to my comments , I’ve conducted the following assessment on each comment previously made:

Comment 1. “The references list seems rather brief…”

Implemented -> suggestion was satisfactorily addressed. The revised version includes additional citations and broader theoretical discussion, it expanded a few sections with richer literature on national image and regional differences.

Comment 2. “I suggest that the authors give more details on how “national image” is defined and measured by using the GDELT database…”

Implemented -> clear and well-argued improvements on section Section 2 and 3 now contain integrated explanations of GDELT metrics and their rationale which makes the methodology more transparent.

Comment 3. “The statistical models are detailed, but it seems to me that some results (for example the insignificance of variables like sister city relationships ) could be more explained…”

Implemented -> the explanation was added. Section 3.1.2 of the revised version includes a discussion on the limited public/media visibility and localized nature of sister city initiatives.

Comment 4. “I suggest that the authors could add more descriptive captions and explanations for figures 1-3…”

Implemented -> yes. Figure 1, for example, includes enhanced commentary and context. The use of AvgTone is described in section 3.1.1 and earlier.

Comment 5. “The authors generalize the perception of Southeast Asian countries as a whole, but I believe they could highlight a little bit more the differences between individual countries…”

Implemented -> with a nuanced and more grounded analysis. Paragraphs in section 2.1 now differentiate countries like Vietnam and Cambodia, citing geopolitical and historical factors.

Comment 6. “I believe it would be helpful if the authors could include a clear “Discussion” section…”

Implemented -> a new "5. Discussion" section has been inserted in the revised version.

Comment 7. “The final section called “Countermeasures and recommendations” section is very insightful but it mixes conclusions with policy suggestions. Perhaps the authors could insert a standalone conclusion…”

Implemented -> the new “6.Conclusion” section has been inserted and summarizes contributions, key results, implications.

Comment 8. “The authors seem to use GDELT data effectively and they could improve transparency by sharing their processed data and scripts (in case they are not confidential)…”

The authors stated willingness to share upon request post-publication.

Only partially implemented -> in revised manuscript the authors provide detailed descriptions of the data sources (GDELT, Comtrade, SIPRI), the methodology used to process them. But, they do not include a direct link, repository, DOI for the processed dataset or the Python scripts used for cleaning, aggregation, analysis. Readers are invited to contact the corresponding author after publication.

Recommendation -> I still suggest that sharing data and scripts via a public platform (like for example GitHub, Zenodo OSF or others) could improve the study's reproducibility and its value.

7. PLOS authors have the option to publish the peer review history of their article (what does this mean? ). If published, this will include your full peer review and any attached files.

**Do you want your identity to be public for this peer review?** For information about this choice, including consent withdrawal, please see our Privacy Policy .

Reviewer #1: No

Reviewer #2: No

---

## [Editor Report · Acceptance letter]

PONE-D-24-56411R1

PLOS ONE

Dear Dr. CHEN,

I'm pleased to inform you that your manuscript has been deemed suitable for publication in PLOS ONE. Congratulations! Your manuscript is now being handed over to our production team.

Kind regards,

on behalf of

Dr. Vincenzo Basile

Academic Editor

PLOS ONE